# Working like a Dog: Exploring the Role of a Therapy Dog in Clinical Exercise Physiology Practice

**DOI:** 10.3390/ani12101237

**Published:** 2022-05-11

**Authors:** Melainie Cameron, Emily Hewitt, Elizabeth Hollitt, Jacqueline Wood, Samantha Brown

**Affiliations:** 1School of Health and Medical Sciences, University of Southern Queensland, Salisbury Rd., Ipswich, QLD 4305, Australia; emilyhewitt050@gmail.com (E.H.); liz.hollitt@hotmail.com (E.H.); jacquie.wood93@gmail.com (J.W.); 2Centre for Health Research, University of Southern Queensland, Salisbury Rd., Ipswich, QLD 4305, Australia; samantha.brown@usq.edu.au; 3PhASRec (Physical Activity, Sport, and Recreation), North-West University, Potchefstroom 2520, South Africa; 4School of Psychology and Wellbeing, University of Southern Queensland, Salisbury Rd., Ipswich, QLD 4305, Australia

**Keywords:** canine, animal-assisted therapy

## Abstract

**Simple Summary:**

Animals may be included in clinical care because they are suggested to reduce patients’ anxiety and stress and increase their interaction with practitioners. In this study, we surveyed and interviewed clients and students from a university teaching clinic about the inclusion of a therapy dog in the delivery of exercise physiology services. The dog and her handler worked together as a team in the clinic for one day a week for most of a year. No other animals worked in the clinic. Services in the clinic were led by students, who were supervised by the exercise physiologist who was also the canine handler. All clients came to the clinic to engage in exercise as medicine, that is, exercise as part of disease management or for recovery from injury or illness. One person who took part in the study had not actually met the dog, so we excluded those results. Everyone else who was surveyed or interviewed considered the dog, a female black Labrador, to be a well-behaved dog, and no-one considered her to be problematic to their engagement in exercise. Including a dog in clinical exercise practice may enhance some parts of the service delivery.

**Abstract:**

Therapy animals in clinical settings are purported to reduce patients’ anxiety, decrease agitated behaviour, serve as social mediators, enhance the social atmosphere, and increase patients’ openness towards practitioners. A therapy dog worked alongside her exercise physiologist handler for approximately 1 day/week in a university clinic. The canine and handler functioned as a team, while the handler simultaneously undertook supervision of students. The clinic was open 24 h/week, and no other therapeutic animal was present for any part of the week. We explored, via surveys and interviews, human responses to the dog. The survey comprised 15 statement items regarding the canine’s role, behaviour, and acceptability in the clinic, ranked from strongly disagree (−2) to strongly agree (2), followed by an open item inviting participants to follow up interviews. Eleven (11) clinical clients and seven (7) students completed the survey. One client had not encountered the canine; these data were excluded. Four (4) participants from the client sample provided subsequent telephone interviews. All participants identified the canine as well-behaved; no participants considered that she detracted from their exercise sessions. Most participants were equivocal to statements regarding social lubrication and openness to practitioners; only three clients and two students identified that they felt more willing to share health information; three students identified that they felt they could confide more in the canine than in the practitioner. Interviewees’ reports were similarly favourable, reinforcing the information obtained from the surveys. Interview transcripts were subject to thematic analysis, which focussed around four key themes: (1) the canine’s good behaviour, (2) clients giving permission, and the canine as both (3) a pleasant distraction from the effort of exercise, and (4) nice to have. A therapy dog may enhance some aspects of exercise physiology service delivery.

## 1. Introduction

Animal-assisted therapy (AAT) is clinical care delivered with the assistance of one or more animals [1]. The mutually beneficial relationship between humans and animals is referred to as a human–animal bond, and is a fundamental underpinning for including animals in human clinical work. 

As well as in emergency medicine, general medicine, and surgical recovery, AAT is used in physical rehabilitation and talking therapies, across a wide spectrum of clinical conditions, and in a variety of clinical settings including residential aged care and hospitals [2].

Therapy animals in clinical settings are purported to reduce patients’ anxiety, decrease agitated behaviour, serve as social mediators, enhance the social atmosphere and increase conversation, allowing patients to be more open with their healthcare providers [1]. Dogs, the animals most commonly engaged in therapeutic settings [2], may enhance the social atmosphere and increase the frequency of conversation between people. This effect, called social catalysis, is somewhat influenced by dog breed [3]. 

There have been several systematic reviews of clinical trials of AAT demonstrating benefits and enhancements beyond usual care. Charry-Sanchez and colleagues conducted a systematic review of 23 trials of AAT (any animals, 13 studies included therapy dogs) in adults with depression, dementia, multiple sclerosis, PTSD, stroke, spinal cord injury, and schizophrenia. Frequently identified outcomes with canine-assisted therapy included reduced loneliness, agitation, and apathy, as well as increased social engagement and enhanced mood [4]. 

Jones and colleagues undertook a systematic review of canine-assisted psychotherapy in adolescents. The included studies provide some emerging evidence of improved efficacy of mental health treatments in self-selected adolescent populations through reducing primary symptoms (anxiety, depression, anger), and secondary factors such as improved engagement, disclosure, and retention in therapy [5].

Coakley and Mahoney undertook a cohort study of 59 haemodynamically stable, hospitalised, adult patients who received a single pet therapy visit (approximately 10 minutes duration) in addition to usual hospital care. Systolic and diastolic blood pressure, and pulse rate were marginally, non-significantly, increased after pet therapy visits, and statistically significant reductions were observed in respiratory rate and patients’ self-reports of pain, along with improvements in energy and mood states. Statistically significant improvements were reported on the tension/anxiety, anger/hostility, fatigue/inertia, and depression/dejection subscales of the Profile of Mood States (POMS), and clinically important changes were noted in tension/anxiety, fatigue/inertia, and total mood disturbance [6]. 

Post-operatively, a therapy dog may assist in children’s recovery from general anaesthesia and surgery, as demonstrated by more rapid return of beta activity on ECG (*p* < 0.001), and reduced pain perception (*p* = 0.01), when compared with usual recovery procedures among controls [7]. Similarly, children diagnosed with leukaemia or solid tumours may obtain notable reductions in pain (*p* = 0.046, d = −0.894) and stress (*p* = 0.005, d = −1.404) from three thirty-minute sessions of structured canine engagement [8].

Rodrigo-Claverol and colleagues incorporated AAT into existing physiotherapy sessions among institutionalised older adults with cognitive impairment. Statistically significant improvements were found in both groups; however, comparing the usual care and adjunctive AAT groups post-intervention, and adjusting the post-intervention scores for baseline measures, the AAT group showed significant additional improvement in communication (Holden scale) [9]. 

There is a considerable body of evidence regarding the use of therapy dogs in specific settings, including emergency departments, palliative care and hospices, secure dementia units, schools, universities, psychiatric services, residential aged care environments, and prisons [10]. Despite the variety of settings in which we find therapy dogs, and the evidence that they can be a beneficial adjunct to clinical care, perceptions of the value and role of therapy dogs varies among clinicians and patients. Moreira and colleagues explored the attitudes and beliefs of nurses and guardians of children and teenagers with cancer towards therapy assisted by dogs, and identified widespread misunderstanding that dogs were included in the oncology service merely for distraction and entertainment, rather than for clinically therapeutic purposes. Both groups of participants identified behavioural changes in the children after therapy dog visits, and nurses identified that the presence of dogs increased their own rapport and connection with child patients, but these changes in social connection were not necessarily seen as having therapeutic value [11]. 

Abrahamson and colleagues explored the views of hospital staff members (nurses, volunteers, administrators) regarding therapy dogs in a medium-sized hospital. Perceptions were largely positive, including identifying that although therapy dogs entered the hospital to provide AAT for patients, their presence also served to reduce staff stress. Comfort, distraction, and social lubrication with patients were also identified as key benefits. Some participants had reservations about dogs in clinical settings: practical concerns, such as infection control, and adequate training for canine and handler, were noted. Some drawbacks, including the possibility that dogs may sadden patients who missed their own dogs, were purported, although there were no examples that this actually occurred [12].

In this study, we aimed to explore the role and acceptability of a therapy dog in an exercise physiology practice on a university campus, a new setting and service for AAT. There appear to have been no previous studies of canine-assisted therapy within an exercise physiology service, and very few studies undertaken in university ambulatory care clinics. Most studies have been undertaken in hospitals and residential care settings, and some in private practice settings. Our goals regarding introducing a therapy dog to a student-led university exercise physiology clinic were to: (a) introduce students to AAT, and (b) explore the possible ways that AAT might enhance exercise physiology practice. 

## 2. Materials and Methods

Across the course of one academic year, Bella, a female black Labrador, trained and accredited as a therapy dog, and her accredited exercise physiologist handler, worked for up to 8 h (1 day) per week in a university teaching clinic. Bella and the exercise physiologist functioned as a canine–handler clinical team (accredited level 1, Therapy Dogs Australia), while the handler simultaneously undertook student supervision. Exercise physiology students worked with adult clients with chronic, complex health conditions. Clients engaged in exercise for the management of clinical conditions known to benefit from regular exercise. Clients had diverse goals, and exercise was prescribed for a range of clinical purposes. Student exercise physiologists oversaw clients’ exercise for client safety, to maintain motivation, and to ensure that exercise load was adjusted as needed. The clinic was open to the public for 24 h each week, and no other therapeutic animal was present for any part of the week. 

When commencing a clinical training placement that included Bella, students were provided with rules and expectation for working with a therapy animal, and some background reading to inform their clinical practice. Involvement of Bella in the clinical setting was entirely optional. No students were compelled to work with Bella, and no clients were exposed to Bella without their voluntary, informed consent. 

Where possible, students engaged Bella in their clinical service delivery, including her in walking groups, inviting clients to bend or move to the ground to pet to her, playing ball games with Bella, allowing Bella to accompany distressed clients through difficult moments such as symptom-provoking tests, and using Bella as an obstacle in some mobility activities. The provision of AAT was not universal; only where AAT aligned with the clinical goals for a given client was Bella directly included in their care. Although Bella was not purposefully included in the provision of exercise services with all clients on a given day, she was present in the clinic for the whole day and available to all clients and students to interact with her if they wished. When Bella was engaged in AAT with a client, the purpose, goals, and process of this therapy were noted in the client’s file.

Bella worked freely, off leash, in all indoor areas of the clinic, and outdoors in a veranda and adjacent garden. Bella was leashed when moving around any other areas of the university campus. She was provided with a mat in a quiet space indoors so that she could retreat, rest, or sleep, and a water bowl with fresh water was available to her on the veranda. The garden area adjacent to the veranda was available for Bella’s toileting. Consultations in the university clinic are typically an hour duration. Bella was engaged with clients for approximately 5–15 min of a consultation, and was free to move to her mat to rest between engagements. Bella was able to exit the clinic to toilet and to drink water as needed. Bella and all the students and staff members took a one-hour break for lunch in the middle of the day.

We explored, via surveys and interviews, the function and acceptability of a therapy dog in this setting. 

### 2.1. Participants

All clients who attended the clinic, and all students who were rostered to placement in the clinic, across the year, were invited to participate in this project.

### 2.2. Measures

Two versions of a survey, one for use with clients, and the other for use with students, were developed for this project. Surveys were piloted with potential participants for face and content validity, and some refinements were made in response to the feedback received. The finalised surveys each comprised 15 items, which included statements regarding the therapy dog’s role, behaviour, and acceptability in the clinic setting, ranked from strongly disagree (−2) to strongly agree (2).

Sample items included: I like Bella; Bella is a well-behaved dog; Bella’s presence in the clinic detracts from my exercise session; I feel as though I can share more with Bella than I can with the students and staff in the clinic; Bella’s presence makes me comfortable to provide information about myself to clinic staff and students; Bella and I share similarities (e.g., age, osteoarthritis, feeling better when I am of healthy weight). Some survey items were intentionally reverse biased to allow us to identify “blank” responses. 

The survey did not cover any clinical information. This study was exploratory of the acceptability of a therapy dog in this setting, and survey items focussed only on this topic. An open item at the conclusion of the survey invited participants to follow-up interviews if they wished to express further views. Interviews were semi-structured, using the survey as a framework for open discussion. 

### 2.3. Procedures

Procedures for this research were approved by the University of Southern Queensland (USQ) Human Research Ethics Committee (H19REA267). 

The project was promoted via posters in the USQ Sport and Exercise Clinic. Copies of the survey were made available in the clinic reception area, and completed surveys were returned to a sealed box in the clinic waiting room. Surveys were provided in hard copy to clients upon attendance to the clinic. Hard-copy data collection occurred over 4 weeks, and was ceased when the clinic closed its face-to-face services during the COVID-19 pandemic.

Survey responses were anonymous unless participants opted to identify themselves. Returned surveys were hand-coded (−2 to 2, with 0 representing a neutral response). 

Participants who identified themselves were contacted for an interview with one of the researchers to further explain their survey responses and provide more information about their experience with Bella. Interviews were undertaken as internet telephone calls. Each interview was audio-recorded, and transcribed verbatim prior to content analysis. 

### 2.4. Statistical Analysis/Analysis

Survey data were analysed descriptively, and reported as mean and mode and standard deviation. 

Interview transcripts were subject to thematic analysis using a common six-step process: familiarization, coding, generating themes, reviewing themes, defining and naming themes, and writing up [13]. To ensure inter-rater reliability, all steps were undertaken by two members of the research team acting independently. 

## 3. Results

Eleven (11) clinical clients and seven (7) students completed the survey. One client who completed the survey had not actually encountered the therapy dog—these data have been excluded (see Table 1). Four (4) participants from the client sample provided subsequent telephone interviews. 

Because the study was exploratory, we simply aimed to document students’ and clients’ views and attitudes towards the therapy dog. Survey results were generally favourable of the inclusion of a therapy dog in the clinical setting. All participants identified the therapy dog as well-behaved, and no participants considered that the therapy dog detracted from their exercise sessions. Participants were somewhat equivocal to statements regarding social lubrication and openness to healthcare providers; only three clients and two students identified that they felt more willing to share health information because of the therapy dog, and three students identified that they felt they could confide more in the therapy dog than in the clinical staff. Summary data of survey results are provided in Table 1. 

Interviewees were similarly favourable and supportive of the inclusion of a therapy dog in an exercise physiology setting. Four key themes arose from the interviews: 

### 3.1. Behaviour

Participants reported that the canine was a well-behaved dog and that her behaviour positively influenced their feelings. They consistently described her as “a good dog”, “calm”, “soothing” and “quiet”.


*Well, firstly, I enjoy dogs. I have a dog myself. And I find dogs calming and soothing anyway, not mad dogs, but Bella’s certainly very quiet and soothing.*
(Interviewee A)

Participants read meaning into Bella’s behaviour, including that Bella understood the needs of clients in the clinic, and held things in common with them. 


*I think she obviously knows what’s going on, and who’s doing what, and how everyone’s feeling because like I was quite happy, and she knew that. She she’d wag her tail and what have you, and I’d talk to her. And you could see some people you know that she’d go over to they’d be, you know, bit iffy. However, she soon settled them down. … you could tell what she was doing… she was assisting everyone that was walking in the door.*
(Interviewee C)


*Sometimes she was maybe a bit antisocial and wouldn’t come up to you, but maybe she was in pain because I’ve got terrible arthritis I know how she feels… It was as if she couldn’t get up. If you went over to her it was fine but it hurt to get up so.*
(Interviewee B)

### 3.2. Permission

Consent (permission) is integral to clinical service delivery, however, participants identified for us that they were asked specifically for their permission for the inclusion of a dog in therapy. Participants also reported that the canine was obedient of her handler and of them. She only approached participants with their permission, and they were able to refuse her involvement in any or all of their consultations. 


*It was all explained very well why she was there. And we were asked to give our permission too.*
(Interviewee A)

Additionally, on the theme of permission, one participant told two vignettes of people she knew who were unwell, one permitted to have a dog in a clinical care setting and the other not so. 

*Just as an aside, the lady that used to live opposite me… she died of cancer, ovarian cancer. And she had to go up to her daughters in [town] and her sister had a dog, Toby, and before she died she was allowed to take it into the care home.… And the dog sat on the bed and she had a pat of her beautiful dog…. And a neighbour here has a mother in care. And he’s got mother’s dog. All mother wants to do is see her dog. She’s not allowed. She can look at it through the window. That’s not going to help her at all. She needs to be able to touch that creature she adores*.(Interviewee B)

### 3.3. Distraction

Participants reported that the canine served as a pleasurable distraction from chronic and complex disease, and the effort of clinical exercise. Participants primarily described that Bella’s presence helped them relax and relieved some of their anxiety. 


*It’s good because you’re sitting near, you’ve got the dog there and give her a rub up… and it just takes your mind off everything and just you’re just looking at the dog and thinking about the dog and the good thing she is.*
(Interviewee C)


*She makes you happy… Relaxes you. All those things. You give her a pat. She loves it and you love it too.*
(Interviewee B)

*I think it actually took my mind off the pain of exercise… A distraction because I’d be thinking about the dog and or looking at the dog and just a sweet, calm dog. So I think anything that that gives you any motivation or any help while you’re exercising, is a good thing. Very good thing*.(Interviewee A)


*It reminds me of when I had a dog and it just makes me remember all the fun and all the friendly you know feelings that I had when we had our own dog.*
(Interviewee D)


*Sort of calms you down when you’re feeling nervous, anxious. You see the dog wagging its tail at you and you just get you know get a real calm feeling.*
(Interviewee D)

### 3.4. Nice to Have

Participants consistently described the canine as a pleasant addition to the services of the university clinic, but not part of the core business. 


*And if Bella isn’t there, does it affect you in any way?*
(Interviewee)


*No, not at all. It’s just nice. It’s nice to have her there.*
(Interviewee A)


*Um, well, you know, she’s not there. So a sort of, there’s a little gap there of nothing.*
(Interviewee C)


*Oh I miss her but it doesn’t affect me, it doesn’t affect me, no. I can still exercise and do what I have to do but I do look for her and think “where is she?”*
(Interviewee D)

## 4. Discussion

This study was exploratory, probing the acceptability of including a therapy dog in a new clinical setting, namely a student-led exercise physiology service within a university. Survey results from students and clinical clients showed generally favourable views towards the inclusion of a therapy dog in this service. When interviewed, clinical clients offered further insights that aligned with the responses to the survey. 

Bella was universally reported as a well-behaved dog (100% of survey respondents strongly agreed with this item), and described as likable with a soothing, calming presence in the clinic (“behaviour” theme). Behaviour and temperament of an animal are integral to the success or failure of AAT; a misbehaving dog may be a problem rather than an asset in a clinical setting.

Our particular clinic has predominantly an older adult clientele engaged in exercise to help manage chronic non-communicable diseases. Bella shared some things in common with our clients. Bella was 10 years old, a senior dog, when she first attended the university clinic. She also had some osteoarthritis in the large joints of her hind limbs that limited her capacity for rapid movement. 

Although Bella may have served as a social lubricant among our client group, we acknowledge that a quiet, slow moving, older dog would not necessarily be a good fit for all types of exercise services. Our clients played ball games with Bella, and these games provided opportunities for clients to challenge themselves physically, and move in unaccustomed ways (e.g., bending to the floor to pick up a ball, throwing with their non-dominant arm). The same ball games played with the same dog may have been unsuitable interventions if working with younger, more mobile clients who might throw a ball too far or fast for the dog to fetch—therapy dogs should be well-matched to tasks to minimise stress, miscommunication, anxiety, or disinterest in their role [14]. We would like to expand our work in AAT to include consideration of the relationship and fit between dog temperament and behaviour, and the work required of the dog in the clinical setting. 

Therapeutic exercise requires client engagement, repetition, and, to make physiological gains, progressive increases in training demands. An ongoing challenge in clinical exercise practice is maintaining the client’s adherence to, and enthusiasm for, exercise. “Get a dog” is oft-suggested common wisdom to motivate older adults to undertake regular exercise. Curl and colleagues determined, from a large population-based study, that dog ownership per se is not related to increased physical health or positive health behaviours such as regular exercise [15]. Rather, dog walking is associated with more favourable body mass, fewer limitations in activities of daily living, and fewer doctor visits. Dog walking also appears to correlate with overall exercise behaviour, and unsurprisingly, the more strongly bonded a person is with their dog, the more likely they are to walk that dog. 

Many of our clients live alone, or with one other person. They may experience shrinking social and physical worlds, as their physical limitations reduce their mobility in the community, and their own longevity bites with the double-edged sword of outliving one’s friends. Some clients told us that they were no longer able to have a dog of their own, and appreciated being able to share “ownership” of Bella through the clinic. Because it is not simply owning a dog, but whether you walk the dog that contributes to your health [15], we wanted to probe whether the inclusion of a dog in an exercise physiology clinic might serve as an exercise motivator. 

Participants told us that they noticed Bella’s absences from the clinic, but for the most part, asserted that her absence had no effect on their exercise behaviour (“nice to have” theme). Interviewees described Bella as a pleasant alternate focus from the difficulties of exercise or chronic disease, reducing the anxiety they can feel with clinic visits (“distraction” theme). Students unanimously reported that they liked the idea of including a dog in clinical exercise physiology services. Clients also strongly favoured this item in the survey, and reinforced this message with their comments in interviews. Participants and students told us that they looked forward to their exercise sessions because they knew that Bella would be there, and that they would recommend a clinic with a therapy dog to others. Notably, our survey respondents were largely pro-dog: the modal response among students and clients to survey item 12 (In general, I like dogs.) was 2 (strongly agree).

Our results appear consistent with a study by Reddekopp and colleagues on patients’ views on being visited by a therapy dog in the emergency department (ED) of a hospital [16]. Most patients (80%) responded favourably to the idea of dog visits in the ED, and current dog owners were overwhelmingly supportive (95%). The majority of patients identified the potential value of dog visits in this setting in terms of reducing anxiety and frustration, or increasing comfort or satisfaction (87% to 92% across these items). Far fewer patients identified the potential for pain reduction as a possible benefit of AAT (59%). In both this study and our own, the substantive clinical potential of AAT (pain reduction, exercise adherence) is overlooked by participants in favour of the comfort and enjoyment dogs provide. 

Because of the anonymous nature of our survey, we could not relate any participants’ responses with clinical attendance data, however, we noticed that some clients’ behaviours suggested a bond had been formed with Bella—they greeted Bella first upon arriving at the clinic, brought her gifts, and asked us about her when she was absent. Clients dropping out of exercise (i.e., non-adherence) is a particular challenge in exercise physiology practice. In future studies, we would like to explore whether AAT influences adherence to clinical exercise, perhaps by increasing accessibility for people unfamiliar with exercise, or by reducing dropout rates from ongoing training. 

Anthropomorphism—the attribution of human traits, emotions, and intentions to non-humans—has long been unpopular among animal behaviouralists [17], but may be a useful tool in animal-assisted clinical interventions with humans [1], as well as in prompting other desirable human behaviours (e.g., reduced meat consumption [18], environmental conservation [19]). Although most participants were neutral to the survey statement that they had attributes in common with Bella (mean 0.80, mode 0), similarities did exist between Bella and the clinic clientele. Most clients at the university clinic are older adults, among whom arthropathies and other chronic conditions are common, and overweight, overeating, and physical inactivity are frequently identified challenges. Bella, as an older dog, with some large weightbearing joints affected by osteoarthritis, and a typical Labrador-esque food drive, served as a prompt for students to commence difficult conversations with clients about overeating, weight gain, physical inactivity, arthropathies, and joint health. Bella had a history of fearful behaviour when first rehomed, a tale often shared with clients who were lacking confidence in their ability to commence new exercise at an older age. Perhaps it is possible to teach an old dog new tricks, and to teach a little old lady to pump iron. Anthropomorphism was particularly apparent when the clinic manager added Bella’s photograph to the “staff on duty” notice board. 

Social catalysis may be both a benefit and a risk of the human–animal bond. Fine [1] maintains that the human–animal bond drives increased disclosure from clients. That is, they are likely to share more with us than they ever expected. Although not captured in the interviews, Bella’s handler noted several anecdotal accounts of grief and loss from clients during their interactions with Bella. Conversations that began “I once had a dog like Bella…” progressed to “…and then that dog passed away…” which then led on to recollections of other griefs, sometimes followed by a bemused “I am not sure how we got onto that.” 

Disclosure in clinical practice needs to be met by clinician capability; that is, it is important that clinicians have capacity to meet clients’ needs, either in direct care provision, or via referral to other providers. Clinicians’ judicious use of the human–animal bond is key to AAT. Bella’s handler is an experienced clinician (25+ years) with the professional skills and network to respond to a wide range of client needs. We offer a caveat that AAT may not be suitable for novice clinicians—the early years of transition to clinical practice are typified by erratic confidence and somewhat rigid adherence to learned procedures, accompanied by some introspection [20]. The challenges of early clinical practice are substantive [21], and novice clinicians may find themselves overwhelmed and unable to manage the increased disclosure and clinical demand generated by including a therapeutic animal in their work. 

Another avenue for further study could include investigation as to whether inclusion of a dog in clinical exercise physiology service prompts changes in any physiological measures of stress (e.g., blood pressure, heart rate, salivary cortisol). We emphasise that reduction in stress measures is not always desirable in the clinical exercise context. Although clinicians do not intend to distress clients, exercise itself can be used as a physiological stressor, and temporary elevation in heart rate or other measures may be desired outcomes. It is important to determine whether AAT could possibly hamper exercise service delivery by dampening desired physiological responses. 

The university clinic is both a teaching space and a healthcare delivery service. Student clinicians were supported and guided in their clinicial practice (including AAT) by the handler’s clinicial experience. Although we did not gather any qualitative reports from students as part of this study, we observed the enthusiasm with which students embraced Bella’s involvement in the clinic, the delight they showed at her arrival each week, and the diligence with which they upheld the “rules of engagement” with Bella (e.g., no snacks). Further, we noticed that during exam periods, when the university campus was filled with anxious students, Bella appeared to be particularly busy, attending to many students, sitting beside them for pats and cuddles. Therapeutic animals are increasingly used in teaching spaces, with some evidence of benefits, but most studies underpinning this work have been undertaken among children, and are of variable quality. These results are not automatically transferrable to the university sector [22]. Studies among clinicians are also somewhat limited—significant reductions in salivary cortisol (a stress biomarker) were noted in a small group of nurses when they were provided with a break for AAT during stressful work, but the same benefits were not seen in nurses who worked in less stressful environments [23]. Areas for exploration following on from this study could include the effects of a therapeutic animal on young adult student practitioners, exploring whether the inclusion of a canine enhanced or complicated students’ clinical service delivery, and whether there are any measurable therapeutic benefits for university students such as enhancement of learning, cohesion among the student cohort, management of anxiety or stress, or reduction in burnout. 

### 4.1. Limitations

We acknowledge that the samples in this project are small, may be biased in favour of people who like dogs, and that the results may apply only to one canine in one clinic. We conducted survey data collection in hard copy only, which we anticipate reduced our total recruitment. We had intended to distribute surveys online to extend recruitment, however, when the clinic was closed to face-to-face consultations due to the COVID-19 pandemic, we considered that it would be somewhat “tone deaf” to engage clients and students in surveys about a clinic they were unable to attend. 

### 4.2. Recommendations

Further exploratory research could be undertaken to determine the ideal canine attributes for exercise physiology settings (e.g., temperament, activity level, vocalisation, bonding behaviour with humans), and the effect of canines on clients’ exercise adherence, stress or anxiety during clinical exercise, and on student practitioners during clinical service provision. 

We recommend that any clinician interested in AAT should invest in some training for themselves and their dog prior to engaging their dog in the workplace, and comply fully with all workplace health and safety requirements. 

## 5. Conclusions

Animal-assisted interventions are possible within clinical exercise physiology. A therapy dog may enhance some aspects of exercise physiology service delivery, including serving as a social catalyst for client disclosure. A dog may be intentionally and actively engaged in the delivery of clinical exercise interventions.

## Figures and Tables

**Table 1 animals-12-01237-t001:** Summary of responses to survey items.

Item	Students (*n* = 7)			Clients (*n* = 10)			Total (*n* = 17)		
	Mean	Mode	SD	Mean	Mode	SD	Mean	Mode	SD
I like Bella.	2	2	0.00	1.80	2	0.42	1.88	2	0.33
Bella is a well-behaved dog.	2	2	0.00	2.00	2	0.00	2.00	2	0.00
I enjoy having Bella in the clinic.	1.86	2	0.38	1.90	2	0.32	1.88	2	0.33
Bella’s presence in the clinic detracts from my exercise session.	−1.00	−1	0.82	−1.60	−2	0.52	−1.35	−2	0.70
I look forward to my exercise sessions / to being in the clinic because I know that Bella will be present.	1.29	2	0.95	1.20	2	1.14	1.24	2	1.03
Bella’s presence makes no difference to my exercise sessions; I do not miss her if she has the day off.	−0.29	0	0.95	−0.50	0	1.43	−0.41	0	1.23
If I had a choice between two clinics with all things equal, I would choose to come to USQ because of Bella.	1.43	2	0.79	0.60	0	1.35	0.94	2	1.20
I feel as though I can share more with Bella than I can with the students and staff in the clinic.	0.57	0	0.79	−0.56	0	0.73	−0.06	0	0.93
Bella’s presence makes me feel comfortable to provide information about myself to clinic staff and students.	0.43	0	0.79	0.22	0	1.30	0.31	0	1.08
I formed a relationship with Bella faster than I did with the clinic staff and students.	0.43	0	0.79	−0.33	0	0.71	0.00	0	0.82
Bella and I share similarities (e.g., age, osteoarthritis, feeling better when I am of healthy weight).	0.14	0	1.21	0.80	0	0.92	0.53	0	1.07
In general, I like dogs.	1.86	2	0.38	1.40	2	0.97	1.59	2	0.80
I like the idea of using a therapy dog in a clinic setting.	2.00	2	0.00	1.60	2	0.52	1.76	2	0.44
If I had a choice between two exercise clinics with all other things being equal, I would choose the clinic with the therapy dog.	1.71	2	0.49	0.90	2	1.20	1.24	2	1.03
I would recommend a clinic with a therapy dog to a friend.	1.57	2	0.79	1.60	2	0.70	1.59	2	0.71

Note: Response range from strongly agree (2) through neutral (0) to strongly disagree (−2).

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
