# Peer review of "Working like a Dog: Exploring the Role of a Therapy Dog in Clinical Exercise Physiology Practice"

_animals, 2022, doi:10.3390/ani12101237_

Round 1

Reviewer 1 Report

Thank you for allowing me the opportunity to review this novel article. Please find my comments below. 

Introduction:

Line 53: I am unsure what you mean by communication therapy. Based on the mental health focus of your literature review I think you are discussing ‘talk therapy’ as in psychology, psychotherapy, counselling, however communication therapy is more frequently considered intervention on communication e.g., speech therapy.

Line 73: There appears to be two separate sentences in this one sentence, and I am bit unsure how it follows on from the previous thought.

Line 139: What was the population of these “hospitalised patients”? Were they children, adults, adolescents? Were they pre or post-surgery? Did the article report this information?  

Line 197: I am unsure how the following findings suggest “reservations about dogs in clinical settings” it seems more as though they perceived different benefits to the authors.

Line 216: I am unsure from reading your introduction what the justification is for your study? What is the current gap in the literature and how did you plan to fill this gap with your research?

Materials and methods:

Line 231: How many hours was the therapy dog engaged in therapy on average?

Measures: Was the survey tested prior to distribution?

Line 293: Is there a reference for the analysis process that you followed for the interview data?

I am unsure why a survey was chosen with such a small sample size. Were you guided by any literature on this?

Similarly, the participant numbers for the interview part of this study is also low. Is there a reason you were unbale to continue to collect data e.g., do another block of therapy with new students and clients?

Results

As I am unclear of your aim: What exactly you were hoping to find out and why, I am unsure exactly what the results are telling me. In your discussion you identify the acceptability of the program, however I am unsure how the themes inform this question.

Discussion

Line 375: Is there a reference for this statement?

Line 400: I am unsure why this information makes her suited to your particular client population, is there any research to reference here?

Paragraph starting 405: Is there any references to support this discussion?

Paragraph starting 430: Walking a dog is a caring/nurturing behaviour. Literature has found that caring behaviours are increased when there is a stronger bond between the individual and the client. Where any questions asked to understand the body that the clients had formed with the therapy dog that could have influenced this? Was the exercise pitched to the client as being a benefit to them or a way of helping the therapy dog?

Line 449: How did you assess this bond that was formed?

Line 488: I am unsure how this paragraph fits into your article.

Line 525: Is there a reason why you question the impact on stress. Was this a theme that was frequently brought up by participants?

Line 540: I am unsure if this paragraph can be included as it was not included in your study, results and appears to be a subjective view of the handler therefore has the potential for bias.

Author Response

Additional notes to the editor: 

Please note that considerable detail was removed from the Introduction in the first review of this manuscript when we were directed to condense and focus our review of literature. We are concerned that in this second revision the reviewers appear to be asking us to add that detail back in.

Also note that Reviewer 1 has reviewed the manuscript with no mark up, and Reviewer 2 has reviewed the manuscript with the tracked changes shown. This means that the line numbers differ between reviews. 

Reviewer 2 Report

Dear Authors,
your study is interesting and well structured. As an exploratory study, it needs further investigation and modifications, but I am sure it will meet the approval of the Experts in this area.

Line 1:  Please, remove "Original research".

Lines 2-3: The title is interesting but its beginning "Gone to the Dogs" is unclear and/or open to free interpretation. I agree with the need to make the title of a manuscript more attractive but it would be appropriate to do it in a scientific way. Please modify.

Line 44 - Keywords: Please remove "therapy dog" and "exercise physiology" that are already cited in the title.

Line 118: Please modify "ATT" in "AAT"

Line 435 below - References: Please check the formatting of the references inserted as they are not consistent with the Instructions for the authors of the Journal.

Round 2

Reviewer 1 Report

Thank you for the opportunity to again review your article. I hope you find the below comments helpful. 

Introduction

Line 57: Reference?

Line 60: The final statement about the dog’s breed seems out of place. How does the dogs breed influence the reported outcomes?

I can see you have identified the gap in the literature regarding the inclusion of a therapy dog into a University Exercise Physiology clinic, however it remains unclear within your introduction why this should be an area of further exploration. For example, beyond the general literature supporting the benefits of animal-assisted therapy, what does the literature say about AAT in exercise physiology specifically? Has there been any studies completed about AAT being included into student clinics for other health professions? What has the literature found here? A student clinic has specific goals, what was your team’s goal for including animal-assisted therapy into your student clinic? Was it specifically for the clients or was it for the student learning experience?   

Materials and method

Line 126: I can see that you have stated that she worked to up to 8-hours (1-day) per week, however it is important to be clear from an animal welfare perspective how long the therapy dog (on average) was engaged in therapy e.g., there would have been times that she was provided with rest breaks, toilet breaks, was not required to specifically interact with a client so would be allowed to rest.

Measures

Line 163: I can see you have stated that the surveys were piloted for face and content validity prior to distribution to participants but it is not clear how this was completed. Was it done with the authors only? Were changes made in between each review?

Line 172: Why were some questions intentionally revers biased?

Results

I am still confused about the theme permission as the quote used appeared to be discussing the process as a whole i.e., being provided with information about the sessions and providing consent, however the description of this theme focuses on the therapy dog listening to the handler and responding to the client’s cues. I am unsure how this theme relates to the acceptability of the program.

Discussion

Line 298: “Get a dog” is often suggested as a motivator for regular exercise for older adults. Is this referenced or the opinion of the authors?

Line 333: Perhaps “demonstrated behaviours suggesting a bond had been formed” will assist you to remain objective.

Line 346: I am unsure of the relevance of the point made regarding the potential for anthropomorphism reducing meat consumption and increasing environmental conservation.  

Line 348: The conclusions drawn from this section still do not appear to be supported by your results. There is one quote made within your results section that states “I've got terrible arthritis I know how she feels... It was as if she couldn’t get up. If you went over to her it was fine but it hurt to get up so” however conclusions made regarding her food drive and fearful behaviours appear to be subjective.

As a whole the discussion appears to consist of mostly the subjective reflections of the author compared to a reflection of the results in line with current literature.

Round 3

Reviewer 1 Report

Thank you for addressing all previous comments. 

This manuscript is a resubmission of an earlier submission. The following is a list of the peer review reports and author responses from that submission.

Round 1

Reviewer 1 Report

Overall comment:

Thank you for the opportunity to read your study. This is a very interesting concept and a study that is needed in both the fields of student education and animal-assisted therapy. However due to the lack of clear aim and, inconsistencies within the interpretation of your results I am unsure if your study fills these gaps. This study would require major revisions to be suitable for publication. 

Please also review the referencing to ensure it is consistent and uses the most up-to-date sources. 

Simple Summary

Line 9 – wording “animals are expected” is incorrect, they have been “suggested”, however the literature is still emerging.

Introduction

There is a large amount of referencing that appears to be missing throughout the introduction.

Line  46 – it would be good to begin with being clear what you are talking about when discussing AAT, as you have a sentence early on discussing the social lubricant effect being dog specific.

Line 51 – I am unsure what you are trying to convey with the following statement “One of the oldest, most convincing studies of this construct explored whether dogs, com- 51 pared with other accompaniments, facilitate otherwise unprovoked social responses, and 52 investigated whether this social catalysis effect is generic or influenced by the type of dog. “ – I am unsure why you have chosen to discuss this study as it is not AAT. There are numerous studies that discuss the social lubricant/social catalyst effect within AAT.

Line 64 – I am unsure of the benefit this paragraph to AAT. Be cautious of using the words “animals included in clinical work serve to enhance therapy”. Animals don’t serve.

Line 71 – there is also a growing body of literature re AAT in physical forms of therapy such as physiotherapy and occupational therapy.

Line 74 – You are discussing why dogs are included in AAT, what other animals are included.

Line 77 – you mention the biopsychosocial model, however, do not expand on this. Unsure why it is included.

Line 80 – I am unsure of the relevance of this study to your topic as it is focusing on mental health.

Line 105 – “of all types of AAT” what do you mean by all types?

The introduction requires a large amount of work to be suitable for publication. An introduction is required to tell the readers a story of what research is currently available in your field, what are the gaps and why your study is needed. This appears to be a summary of some literature of AAT, however, a number of prominent articles that could be relevant to your topic are missing, further it is not clear why your study is focusing on exercise physiology in a university clinic.

Materials and Methods

Line 175, I would recommend using a pseudonym or “therapy dog” for the therapy dog in the study. Is her handler also the owner? What did the eight hours look like? Were they consecutive? How many clients did the therapy dog work with? What exact conditions did the clients have? Where they pain related, psychosocial, mobility? What goals was the therapy dog involved in? Did the clients work with the therapy dog each session that they came? What was the role of the student and what was the role of the therapist? What was the purpose of including a therapy dog into a student led clinic?

Participants: How many were invited?

Measures: What was the goal of the survey? What clinical questions were you wanting answered?

Results:

Similar to my comment about your methodology, I am unsure what the aim of your study was. What clinical question were you trying to answer? Are the initials included at the end of the quotes the real person’s initials?

Behaviour: I am unsure what you are conveying with the results in this section, the choice of quotes is also unclear.

Permission: I am unsure what this result is attempting to communicate about AAT.

Distraction and Nice to have: similarly, I am unsure what these results were attempting to convey.

Discussion:

Line 298 – I am unsure of the relevance of your opening sentence as this was not discussed at all throughout your article.

Line 307 – again I am unsure of the relevance of this information as the legislation of therapy dogs in Australia is not addressed in any other part of your article

Line 315 – As above, none of your results discuss the importance of having therapy dogs appropriately trained or certified.

Line 324 – I am unsure of the placement of this paragraph; it should be included within methodology.

Line 332 – The interpretations made within this paragraph are not clear within your results section. Same comment for paragraph beginning line 346 and 353.

Line 364 – This may be true, however it is not clear within your results that your clients related to her more due to her age.

Line – 398 – Again this is not clear within your results.

The overall discussion requires a lot of work to be suitable for publication. It appears that the interpretations discussed throughout the discussion is drawn from the biased observations of the handler and not the results. Further, minimal discussion occurs regarding to how these results relate to the current literature and how this study has assisted to fill a gap.

Reviewer 2 Report

First, I commend the authors for conducting this study and sharing the findings of this survey. The field continues to see an expansion of therapy animals in several unique disciplines, including the rise of animals in occupational and physical therapy settings. There is a continued need to document the efficacy, feasibility, and acceptability of AAT in these programs as we see increased expansion. However, I do not believe that this manuscript, in its current form, is suitable for publication due its low ratings of novelty, significance, scientific soundness, and quality of presentation. The introduction and discussion sections in particular will benefit from a re-evaluation of how the text is shaped and presented, and the manuscript as a whole requires shortening and re-writing to be more concise and relevant to the large existing knowledge base in this area. I also encourage authors to consider collecting more data, as a sample size of n=3 for the interviews is quite limited and leads to low generalizability of findings. After these steps, it may be a better fit for a peer-reviewed journal outlet.

My specific comments to authors to improve the manuscript are below:

When considering the text as a whole, the manuscript is very long and could be improved by removing a bulk of the word count and creating more concise paragraphs. In the introduction, there is a diverted focus towards outcome studies of AAT rather than perceptions/acceptability of AAT (Wells study 12 lines of text, Ambrosi study 18 lines of text, e.g., plus several more). However, these studies have different populations, outcomes, settings, and research goals than the current study. A thorough, paragraph-long detailed reporting of a previous study would be more appropriate if researchers were directly replicating it or it had findings that were directly applicable to the current study’s hypotheses or rationale. Rather, it would be more appropriate for this introduction to first start with a definition of a therapy animal/AAT, then an overview of therapy animal benefits (citing systematic reviews/meta-analyses and general trends of findings across studies), followed by the current knowledge on perceptions of therapy animals in different healthcare settings. This will greatly improve the authors’ framing of the research question and knowledge gap.

A major issue in this manuscript is that there is no framing of the research question in the larger field in terms of why this study is needed to fill an important gap in the knowledge base of therapy dogs. Therefore, I rated the manuscript's significance as low. Notably, a large plethora of qualitative studies exist on the acceptability and positive perceptions of therapy animals in many different settings, including universities, hospitals, nursing homes, and rehabilitation clinics (e.g., Zents et al. 2017; Moreira et al. 2016; Abrahamson et al. 2016; Bibbo et al. 2013; Reddekopp et al. 2020; White et al. 2015; Hansen et al. 2017; Wu et al. 2002; Machova et al. 2020; Salette et al. 2021). However, the introduction does not discuss this rather extensive field of knowledge, nor do authors explain or articulate what this study is adding to this knowledge base or what this study does to inform or improve AAT practice or research. Further, while the case is made that more empirical research on outcomes is needed with randomized designs and strong methodology, this study does not gather any outcomes and is based purely on qualitative perceptions/acceptability of a therapy animal in a clinical practice with a small sample size. Therefore, the case needs to be made that this specific qualitative study on perceptions/acceptability of therapy dogs in this clinic is necessary to fill a research gap, and how this adds to the extensive knowledge we already have in the field on therapy dog perceptions by staff/clients/handlers in other clinical settings. Therefore, I also rated this manuscript low in novelty, as it is unclear how this study is adding to the field.

Methods: Overall the methods are thorough and written well. Care should be taken to describe the protections in place for animal welfare of the therapy dog. There appears to be a human subjects ethical review, but not an animal subjects review for this research? (Please state if not) Participant recruitment should be described in detail, including the percentage of participants who were contacted vs participated in the study. Finally, the methods are missing crucial details on how the content analysis of the interview transcripts. What theoretical model was followed? How were themes coded and conceptualized? Was there inter-rater reliability or was coding done by one member of the research team? (etc)

Results: The results are clear and concise. I recommend excluding the client without dog contact from all analyses, including Table 1. It would be helpful to also report standard deviation in Table 1, as well as a column with total (n=17). The sample size for the qualitative data (n=3) is very low, and has severely limited generalizability of findings. Therefore, I rated the scientific soundness of the data as low. 

Discussion: The discussion is also very long, and lacks in concise paragraphs/text that clearly state the findings and situate them into the larger knowledge base of AAT perceptions. The beginning text focuses entirely on irrelevant concepts to the results which I believe could be removed to improve the text (e.g., difference between therapy dogs and assistance dogs, therapy animal’s rearing history). Instead, the discussion should begin with an overall re-stating of the study’s findings, followed by how this replicates other similar findings in the field. There is no mention of how this study compares to, or more importantly, adds to, the existing knowledge of perceptions and attitudes towards therapy dogs from clients and students (e.g., Zents et al. 2017; Moreira et al. 2016; Abrahamson et al. 2016; Bibbo et al. 2013; Reddekopp et al. 2020; White et al. 2015; Hansen et al. 2017; Wu et al. 2002; Machova et al. 2020; Salette et al. 2021). Therefore, combined with the missing literature review in the introduction, I also rated the quality of presentation as low. I encourage the authors to consider a re-write of the discussion to specifically frame how the trends of both the quantitative and qualitative portions of this study align, or do not align, with the current knowledge we have on therapy animal programs in similar clinical settings. In addition, a discussion of how this knowledge contributes to clinical practice/translation and research would be beneficial to further situate the importance of the findings to future AAT programs.